# DISCRETE STATE-ACTION ABSTRACTION VIA THE SUCCESSOR REPRESENTATION

## ABSTRACT

While the difficulty of reinforcement learning problems is typically related to the complexity of their state spaces, Abstraction proposes that solutions often lie in simpler underlying latent spaces. Prior works have focused on learning either a continuous or dense abstraction, or require a human to provide one. Information-dense representations capture features irrelevant for solving tasks, and continuous spaces can struggle to represent discrete objects. In this work we automatically learn a sparse discrete abstraction of the underlying environment. We do so using a simple end-to-end trainable model based on the successor representation and max-entropy regularization. We describe an algorithm to apply our model, named Discrete State-Action Abstraction (DSAA), which computes an action abstraction in the form of temporally extended actions, i.e., Options, to transition between discrete abstract states. Empirically, we demonstrate the effects of different exploration schemes on our resulting abstraction, and show that it is efficient for solving downstream tasks.

## 1 INTRODUCTION

Reinforcement learning (RL) provides a general framework for solving search problems through the formalism of a Markov Decision Process (MDP); yet with that generality, it sacrifices some basic properties one might expect of a search algorithm. In particular, one should not explore a state multiple times, which is why basic search, such as Dijkstra's algorithm, keeps track of the explored frontier. Basic search methods perform well when state and action spaces are small and discrete, but don't necessarily translate to more complex environments, such as large or continuous ones.

Humans have several intuitive ways to efficiently explore in complex scenarios. One is by *abstracting states*, thereby exploring a simpler, more structured model of the environment. For example, consider searching for an exit in a large room using only touch: we would never blindly roam the center, but rather follow the walls. By abstracting states based on the property "can contain exit", we greatly reduce the set of states we have to explore in the first place. Another closely related way we explore efficiently is by *abstracting actions*. Ingrained skills, or temporally extended actions, impose a prior on the types of action sequences that help solve problems. For example, when exploring we don't move forward and then immediately backwards. Randomly choosing the next direction to move is rarely a good idea, and so skills ensure that we explore new environments efficiently. In this work we are concerned with incorporating the intuitive concepts of state abstraction and action abstraction into reinforcement learning.

Our method for state abstraction is based on the *Successor Representation*, which intuitively characterizes states based on "what happens after visiting this state". By learning a *discrete* state abstraction, we take advantage of a simple and natural definition for action abstraction: abstract actions are policies which help the agent navigate between pairs of abstract states (Abel et al., 2020). We motivate our interest in discrete abstractions in two ways. Firstly, many of the decisions an agent must make in the world are discrete and depend on discrete objects or properties. For example, the length of the optimal path out of a room does not change continuously as a function of the number of doors. While we can still model such decision problems using continuous representations, it is known that discrete metrics cannot be perfectly embedded in continuous spaces (Bourgain, 1985), and it has been shown empirically that policies trained in such continuous spaces struggle precisely at points of discontinuity (Tang & Hauser, 2019).

The second motivation is that classical algorithms for planning in discrete spaces are better understood and provide stronger guarantees; in fact, we often deal with continuous spaces by discretizing them with the help of local planners (Kavraki et al., 1996). Our action abstraction is like a local planner, except learned rather than specified ahead of time. Moreover, our discrete abstraction represents an explicit reduction in the size of the state space, in which both the depth and branching factor of future search can be easily controlled. Thus, it is simpler and more efficient to reuse it to navigate the environment.

## 1.1 Contributions

Our main contribution is a novel method to learn a discrete abstraction by partitioning an arbitrary state space from a dataset of transitions which explore that same space. In particular, we cluster states with a similar Successor Representation (SR) as being part of the same abstract state. Intuitively, if the dataset of transitions was generated by some policy, states from which that policy visits similar states are in turn marked as similar. Unlike prior works on the SR (e.g., Machado et al. (2018b); Ramesh et al. (2019)), our approach is end-to-end trainable and uses a comparatively weaker max-entropy regularization. Our neural network model resembles a discrete variational autoencoder, in which an encoder computes the abstraction and a decoder computes the SR.

To demonstrate the effectiveness of our method, we propose an algorithm, *Discrete State-Action Abstraction* (DSAA), which creates a discrete state-action abstraction pair, using *options* for modeling the action abstraction. Since the SR depends on the policy used to generate data, we report the effect of changing the exploration method on the resulting abstraction, in contrast to prior work which has focused on uniform random exploration. We additionally compare DSAA to related works on both discrete and continuous tasks, demonstrating the value of learning a simple reusable representation.

## 2 Background

### 2.1 Reinforcement Learning

We consider the model-free reinforcement learning (RL) problem with an underlying infinite-horizon Markov Decision Process (MDP): $\mathcal{M} = (\mathcal{X}, \mathcal{A}, p, r, \gamma, x_0)$, with state space $\mathcal{X}$, action space $\mathcal{A}$, unknown environment dynamics $p(x' \mid x, a)$ giving the probability of transitioning to state $x'$ having taken action $a$ at state $x$, reward function $r : \mathcal{X} \times \mathcal{A} \to [0, 1]$, discount factor $0 < \gamma < 1$, and initial state $x_0$. Let $x_t \in \mathcal{X}, a_t \in \mathcal{A}$ be the agent's state and action respectively at time $t$, and $\pi(a_t \mid x_t)$ be the agent's policy, determining the probability distribution of agent actions at state $x_t$. Given a policy $\pi$, the expected return (i.e., discounted sum of rewards) the agent would obtain if action $a$ is taken at initial state $x$, is described by the *Q-value function*:

$$Q^\pi(x, a) = \mathbb{E}_{p,\pi} \left[ \sum_{t=0}^\infty \gamma^t r(x_t, a_t) \middle| a_0 = a, x_0 = x \right], \text{ where } a_t \sim \pi(\cdot|x_t) \text{ and } x_{t+1} \sim p(\cdot|x_t, a_t).$$

The agent's goal is to compute a policy $\pi$ which maximizes the expected return from the starting state $x_0$, i.e., $\pi \in \arg \max_{\bar{\pi}} \mathbb{E}_{a \sim \bar{\pi}(\cdot|x)}[Q^{\bar{\pi}}(x_0, a)]$.

Since the environment dynamics $p$ is unknown, a common approach in RL is to iteratively improve an estimate of the Q-value function, while simultaneously exploring the environment using the induced policy. However, we highlight that when the reward function is sparse, such methods suffer from a long and uninformed (unrewarding) random exploration phase.

### 2.2 Options Framework

Sutton et al. (1999) presented the options framework to extend the classic formalism of an MDP to a semi-MDP, in which we can replace primitive single-step actions with temporally extended policies in the form of *options*. An option in an MDP is a 3-tuple $o = (\mathcal{I}_o, \pi_o, \mathcal{T}_o)$, where $\mathcal{I}_o, \mathcal{T}_o \subseteq \mathcal{X}$ are the initiation and termination sets respectively, and $\pi_o$ is a policy that initiates in some state $x_0 \in \mathcal{I}_o$ and terminates in any state in the set $\mathcal{T}_o$.

Intuitively, an option describes a local subproblem of navigating or funneling the agent between regions of the state space. A common approach is to create options so that the termination set of one lies in the initiation set of another, thus allowing for option chaining (Bagaria & Konidaris, 2019). In

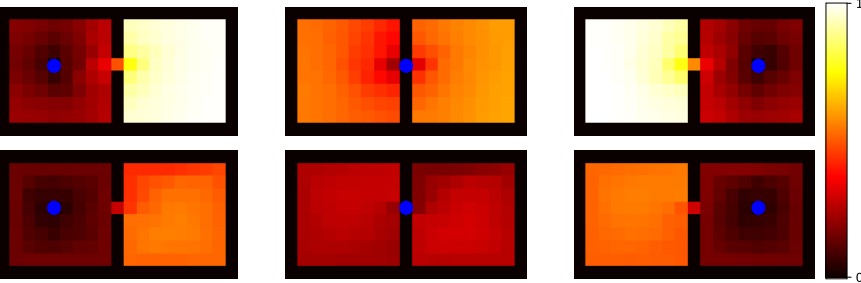

Figure 1: We provide intuition regarding the successor representation in a two room environment with black walls where each room has $8 \times 8$ cells and a single open cell connecting them. We compute the euclidean distance $||\psi(s) - \psi(\bar{s})||$ between the SR of every cell $s$ and the SR of a fixed reference state $\bar{s}$ shown in blue. The top row shows distances computed for a uniform random policy, and the bottom row for a uniform random policy augmented with a wall-hugging option. This option initiates next to any wall, and follows it counterclockwise until there are no walls adjacent. We observe that the SR partitions the environment based on agent dynamics, and options have a large effect on the SR: relative distances are significantly smaller when using the wall-hugging option.

general, the quality of the options depends on the partition of the state space, such that each option takes the agent to a useful sub-goal. *Our work can be viewed as a method for designing such a partition based on the successor representation.*

## 2.3 SUCCESSOR REPRESENTATION

The successor representation (SR) of a state, also known as the discounted state occupancy, models the visitation density of a given policy $\pi$ starting from each state. In a finite discrete state space $\mathcal{S}$ we can simply keep count of these visitation frequencies in a matrix $\Psi_\pi \in \mathbb{R}^{|\mathcal{S}| \times |\mathcal{S}|}$ whose $(u, v)$-th entry is

$$\Psi_\pi(u, v) = \mathbb{E}_{p,\pi} \left[ \sum_{t=1}^\infty \gamma^t \mathbf{1}_{s_t = v} \middle| s_0 = u \right].$$

The SR of state $u$ is the $u$-th row of the matrix $\Psi_\pi$, denoted $\Psi_\pi(u)$.

Much like the Q-value function of an MDP, the SR satisfies a Bellman-like equation which relates the SR of a state to its neighbors by

$$\Psi_\pi(u, v) = \mathbf{1}_{u=v} + \gamma \mathbb{E}_{s \sim p(\cdot|u, \pi(u))} \Psi_\pi(s, v).$$

When the environment dynamics are unknown and we instead have a dataset $\mathcal{D}$ of transitions collected using a policy $\pi$, this equation can be used to learn an estimate of the SR, $\psi : \mathcal{S} \to \mathbb{R}^{|\mathcal{S}|}$ which minimizes the so-called temporal difference (TD) error $\mathbb{E}_{s,s' \sim D} \left[ \psi(s) - (\mathbb{1}(s) + \gamma \psi(s')) \right]^2$ where $\mathbb{1}(s)$ is a one-hot encoding of the discrete state $s$ (the $s$-th standard unit vector).

We highlight that the SR depends heavily on the given policy $\pi$ used to explore the environment. Past work has focused on the setting where $\pi$ is a uniform random policy (i.e., a random walk), and so the properties and/or usefulness of the SR with more general policies remains an open problem. We provide some intuition regarding the effects of changing the underlying policy on the SR in Figure 1. We experimentally explore these effects on our abstraction in Section 6.1.

Prior works have focused on the eigenvectors of the SR, primarily taking advantage of its relation to the graph Laplacian (e.g., Machado et al. (2018b)), and to the decomposition of the MDP value function in terms of the SR (Hansen et al., 2019; Blier et al., 2021). In this work we focus on the SR itself rather than its eigenvectors. Figure 1 demonstrates that the SR acts as a state embedding where distance between states corresponds to similarity between their successors. Moreover, options have predictable effects on the SR; states in an option's initiation set have a more similar SR relative to states outside it. The above observations motivate our method: cluster using the SR as a similarity metric, and then train options whose initiation sets correspond to each cluster.

## 3 RELATED WORK

### 3.1 REPRESENTATION LEARNING

One of the primary motivations for abstraction is that complex real world MDPs have simpler latent structure, and exploring this latent space is sufficient for solving environment tasks (Jiang et al., 2017). For example, two different images may be equivalent with respect to maximizing reward (e.g., the color of an object might be irrelevant for planning). As such, it is desirable to first compute a latent space that preserves underlying structure, and only then solve tasks in that latent space (Lee et al., 2020). One type of abstraction is *Bisimulation*, in which the aim is to preserve properties of the original MDP, namely the value function or optimal policy (Biza et al., 2020; Zhang et al., 2021).

We emphasize that our approach is from the unsupervised learning perspective, in which the aim is to learn a representation without reward, and then transfer this representation to arbitrary reward functions on the same MDP. Many unsupervised representation learning methods attempt to capture MDP dynamics via contrastive losses which bring states separated by short time differences closer together (Stooke et al., 2021; Erraqabi et al., 2021). These methods are said to capture features which change slowly as the agent transitions in the environment, thereby yielding efficient exploration (Li et al., 2021). Jonschkowski & Brock (2015) impose additional structure on the latent space, motivated by real world priors such as Netwon's Laws.

As observed in section 2.3, the SR is another method for capturing long term environment dynamics with respect to an exploration policy. Like us, Ramesh et al. (2019) cluster states based on the SR, yet their method is restricted to discrete inputs, whereas ours learns from arbitrary input spaces in a fully differentiable manner. Also like us, Giannakopoulos et al. (2021) learn a discrete abstraction, but they follow the common paradigm of using observation reconstruction to inform the latent space, which acts as a powerful regularizer encoding information *irrelevant to planning*.

### 3.2 INTRINSIC MOTIVATION

Abstraction falls under the broader category of *Intrinsic Motivation* (IM) methods. IM methods provide an intrinsic reward to the agent to guide and structure the exploration process, which makes exploration more efficient in the absence of environment rewards (i.e., when rewards are sparse). Oudeyer & Kaplan (2009) describe three types of IM: knowledge, competence, and abstraction.

Knowledge-based IM methods reward the agent for improving its knowledge of the environment. Strehl & Littman (2008) and Bellemare et al. (2016) reward the agent for visiting novel states. Curiosity (Sekar et al., 2020) rewards the agent for visiting states in which a trained transition model has high uncertainty, thus encouraging the policy to sample in those regions and improve the model. Competence-based IM methods reward the agent for accomplishing sub-tasks in the environment. They often train a Universal Value Function Approximation (Schaul et al., 2015) and employ strategies such as Hindsight Experience Replay (Andrychowicz et al., 2017) to iteratively improve the agent's competence. These goal conditioned policies often (e.g., Nasiriany et al. (2019)) define goals in a latent space trained from a variational autoencoder (VAE).

Abstraction-based IM methods typically reward the agent for exploring a latent space. For example, Feudal Networks (Vezhnevets et al., 2017) trains a manager worker system, where the manager maps observations to a latent space and produces a direction in that space, then the worker is rewarded for achieving this desired direction. Eigenoption Discovery (Machado et al., 2018b; 2021) computes a deep SR, then uses its eigenvectors as individual reward functions for a set of options. They train the SR offline, using a uniform random exploration policy.

## 4 DISCRETE ABSTRACTION VIA END-TO-END SUCCESSOR LEARNING

In this section we describe our model $M_{\phi,\psi}$ for learning a discrete abstraction given a dataset of environment transitions $\mathcal{D}$. See Figure 2 for a visual representation, and Appendix A.3 for a more detailed derivation of our model loss.

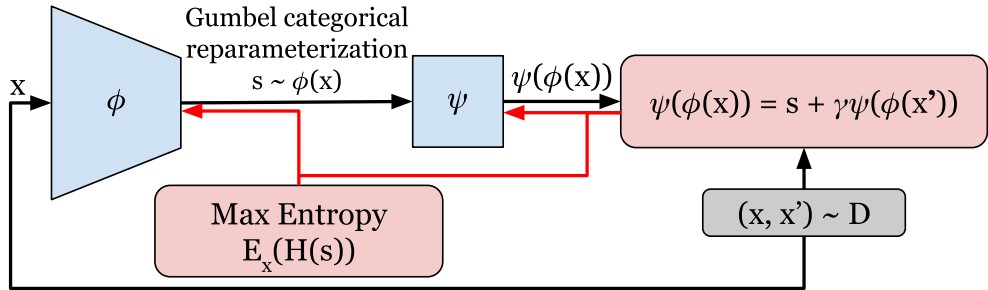

Figure 2: Our model $M_{\phi,\psi}$ resembles a discrete variational autoencoder in which we replace the observation reconstruction decoder with a successor representation module. Red arrows indicate the flow of gradients back through the models. The encoder $\phi$ maps a observations $x$ to discrete abstract states $s$, and $\psi$ learns the SR based on next state samples $x'$ and the TD-error.

## 4.1 ABSTRACTION $\phi$

We implement our discrete abstraction as a feed-forward neural network encoder $\phi : \mathcal{X} \rightarrow \Delta(N)$ which maps observations $x \in \mathcal{X}$ to points on the $N$-simplex (i.e., a distribution over $N$ abstract states). Our model resembles the encoder of a Discrete Variational Autoencoder (D-VAE), where we employ the Gumbel-Softmax (Jang et al., 2017) to sample from the output distribution. Thus, we regularize the distribution, as in a D-VAE, to be similar to a uniform distribution prior $\mathcal{U}(N)$ over $N$ discrete states, by minimizing the KL-Divergence over a dataset of collected samples $\mathcal{D}$,

$$\mathcal{L}_H(\phi; \mathcal{D}) = \mathop{\mathbb{E}}_{x \sim \mathcal{D}} \left[ D_{\mathrm{KL}}(\phi(\cdot \mid x) \,\|\, \mathcal{U}(N)) \right] = \mathop{\mathbb{E}}_{x \sim \mathcal{D}} \left[ -H(\phi(\cdot \mid x)) \right], \tag{1}$$

which is equivalent to maximizing the entropy over $\mathcal{D}$; $H(\phi(x)) = -\sum_i \phi(x)_i \log(\phi(x)_i)$.

Intuitively, we motivate the use of such regularization as follows: a uniform prior encourages a partition of the environment into similar sized parts. Segmenting the state space into abstract states turns our original large MDP into a set of smaller sub-problems restricted to the pre-image of each abstract state. From a divide and conquer perspective, we would prefer these problems to be approximately the same size; otherwise, we may end up with very large abstract states, in which computing the seemingly local options between them is just as hard as solving the original task.

To minimize Eq. 1, the Gumbel-Softmax allows us to sample from the distribution while still back-propagating through it. We qualitatively verify in Appendix A.4 that this is necessary, and that treating $\phi$ as a classifier using simple softmax activation is insufficient.

## 4.2 SUCCESSOR REPRESENTATION $\psi$

The successor representation acts as a *decoder* to the encoder $\phi$. We implement it as a feed-forward neural network $\psi : \Delta(N) \rightarrow \mathbb{R}^N$, and train using the TD error over a dataset of transitions $\mathcal{D}$:

$$\mathcal{L}_{SR}(\psi; \phi, \mathcal{D}) = \mathop{\mathbb{E}}_{x,x' \sim D} \left[ \psi(\phi(x)) - (\phi(x) + \gamma \psi(\phi(x')))^- \right]^2 \tag{2}$$

where $(\cdot)^-$ indicates a fixed target, treated as a constant during back-propagation.

Back-propagating through $\psi$ into the many-to-one $\phi$ ensures that when $\phi$ maps two input states to the same abstract state, they should have similar SR under the induced abstraction. This effectively clusters, much as was done by Ramesh et al. (2019), but does so in a latent space. We note that zero is a fixed point of Eq. 2, meaning $\phi \equiv 0$ and $\psi \equiv 0$ minimizes the loss. As a consequence, prior works tend to compute their abstraction in other ways, e.g., by not back-propagating through $\psi$ (Machado et al., 2018b) or resorting to dense regularization such as observation reconstruction (Kulkarni et al., 2016). Our work shows that with a discrete abstraction, and with Eq. 1 as regularization, we can avoid this degenerate case. In Appendix A.4 we demonstrate that clustering in the latent space learned by a D-VAE decoder yields abstractions not conducive to solving tasks in the environment.

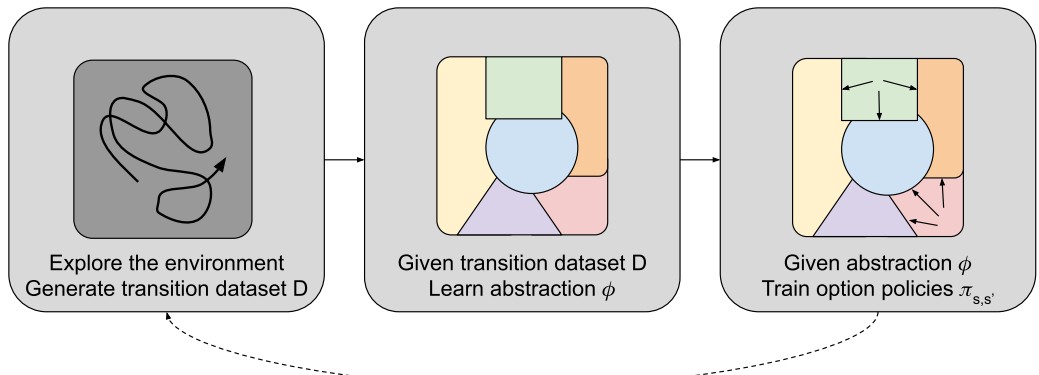

Figure 3: DSAA uses an exploration policy $\pi$ (e.g., explore randomly with a set of options) to generate a dataset of transitions in the environment. Then the successor learning module described in Section 4.1 computes a discrete abstraction of the environment according to $\psi_\pi$, the SR induced by $\pi$. Finally, we train a set of $\phi$-relative options which navigate between pairs of abstract states. These options can be used to solve downstream tasks in the environment, or they can optionally be used by some other exploration scheme to generate new data for a new abstraction.

---

**Algorithm 1** Discrete State-Action Abstraction (DSAA)

---

1: **function** DSAA(integer $N$, transition dataset $\mathcal{D} = \{(x, a, x')_i \mid x' \sim p(\cdot \mid x, a)\}$)
2:     Initialize models $\phi, \psi$                        $\triangleright$ With size $N$ latent space
3:     $\mathcal{L}_A(\phi, \psi; \mathcal{D}) = \mathcal{L}_H(\phi; \mathcal{D}) + \mathcal{L}_{SR}(\psi; \phi, \mathcal{D})$    $\triangleright$ Train abstraction with loss as Eq. 1 + Eq. 2
4:     Update $(\phi, \psi) \in \arg\min \mathcal{L}_A$                  $\triangleright$ e.g., with SGD over $\mathcal{D}$
5:     $G = (V = [N], E = \{\})$               $\triangleright$ Initialize directed abstract graph
6:     $E = \{(\phi(x), \phi(x')) \mid (x, x') \in \mathcal{D}\}$    $\triangleright$ Add an edge for each abstract transition in data
7:     $\mathcal{O}' = \{o_{u,v} \mid (u, v) \in E\}$           $\triangleright$ Train option policy from $\phi^{-1}(u)$ to $\phi^{-1}(v)$
8:     **return** $\phi, \psi, \mathcal{O}'$
9: **end function**    $\triangleright$ Note we can train a new abstraction by exploring with this new set of options

---

## 5   The DSAA Algorithm

We provide pseudocode for our algorithm, Discrete State-Action Abstraction (DSAA), in Algorithm 1 and a pictorial representation in Figure 3.

DSAA takes as input a dataset of transitions in the environment, trains an abstraction $\phi$ as described in section 4.1, and then produces a set of $\phi$-relative options $\mathcal{O}$ (Abel et al., 2020), which navigate from primitive states in one abstract state to another. This induces an abstract graph $G$, in which nodes correspond to abstract states and edges to options between them.

More specifically, for a fixed number of abstract states $N$ and dataset $\mathcal{D}$ our algorithm produces:

1. The abstraction $\phi : \mathcal{X} \to [N]$. Let $\phi^{-1}(s) = \{x \in \mathcal{X} \mid \phi(x) = s\}$.

2. The successor representation model $\psi : [N] \to \mathbb{R}^N$.

3. An abstract MDP represented by the abstract graph $G = (V, E)$. Much like Roderick et al. (2018), for every abstract state transition that occurs in the dataset we add an edge to the graph: $V = [N]$ and $E = \{(\phi(x), \phi(x')) \in [N] \times [N] \mid (x, x') \in \mathcal{D}\}$.

4. Option policies $o_{s,s'} \in \mathcal{O}$, where $o_{s,s'} = (\phi^{-1}(s), \pi_{s,s'}, \mathcal{X} \setminus \phi^{-1}(s))$ for each $(s, s') \in E$. $\pi_{s,s'}$ initiates in the preimage of abstract state $s$ and terminates upon leaving it. $\pi_{s,s'}$ is rewarded for transitioning into $s'$ as follows: $r_{s,s'}(x, a) = \mathbf{1}_{\phi(x')=s'}$, where $x' \sim p(\cdot|x, a)$. We note that this is a slight departure from our previous discussion of options, since we only reward the policy for specific transitions into the terminating set.

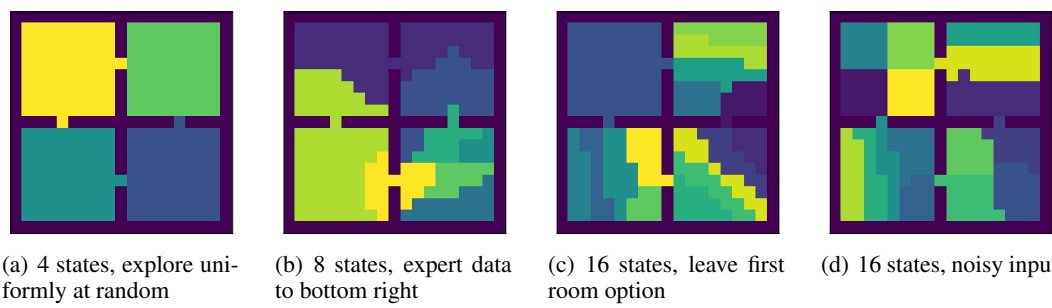

(a) 4 states, explore uniformly at random

(b) 8 states, expert data to bottom right

(c) 16 states, leave first room option

(d) 16 states, noisy input

Figure 4: $M_{\phi,\psi}$ on FourRooms with different exploration datasets. Each color is one abstract state.

### 5.1 USING AN ABSTRACTION TO SOLVE TASKS AND MAXIMIZE REWARDS

From some starting state $x_0 \in \mathcal{X}$, given a *goal state* $x_g \in \mathcal{X}$, we can compute an abstract path in $G$, $P_{x_0,x_g} = [s_0, s_1 \ldots \ldots \ldots, s_k | s_0 = \phi(x_0), s_k = \phi(s_g)]$. To reach $x_g$ we can now execute the corresponding sequence of options, $\{o_{s_i,s_{i+1}}\}_{i=0}^{k-1}$, and finally explore solely within $s_k$. In other words, we have reduced the original exploration problem from the full state space $\mathcal{X}$ to just $\phi^{-1}(\phi(x_g)) \subseteq \mathcal{X}$. We highlight that baseline exploration algorithms specifically struggle with long trajectories through bottleneck states, and as we demonstrated intuitively in Figure 1 and experimentally in Section 6, our SR based abstraction partitions the environment along bottlenecks.

If instead of a goal state in the environment we are given a *reward function* to maximize, we simply explore using the trained options. In other words, we perform a lazy random walk on $G$, in which at every step we select a random neighboring edge and follow the corresponding option policy. The walk is lazy because we allow for self-loop edges, whose options randomly explore without leaving the abstract state. We observe that this approach introduces an exploration-exploitation trade-off: at each step the agent either *explores* within its current abstract state, or *exploits* a trained option to reach a new abstract state. The agent is trained in the standard semi-MDP fashion, where we take sums of rewards over temporally extended actions (Sutton et al., 1999)).

## 6 EXPERIMENTS

### 6.1 EFFECT OF CHANGING THE DATASET

In these first experiments we explore the effects of changing the input dataset on our model $M_{\phi,\psi}$. We report results in Figure 4 on the FourRooms environment (Sutton et al., 1999) using (a) uniform random exploration, (b) expert data that takes the agent to the bottom right corner, (c) random exploration with two options which exit the first room, and (d) random exploration with noisy input, where each state has a randomly sampled noisy bit that changes each time the state is visited making different visits seem different. In each case we train with the same parameters (e.g., model size or loss hyper-parameters) and only vary the number of abstract states $N$. Figure 4(a) demonstrates that our abstraction learns the intuitive partition under random exploration. Figures 4(b) and 4(c) show how the initiation sets of options (including expert policies) get grouped together. Finally, Figure 4(d) demonstrates our model's robustness to irrelevant features, as it successfully captures environment dynamics (e.g., state proximity) despite the existence of distracting features.

### 6.2 COMPARISON TO RELATED WORKS

In these experiments we compare to two related works, Contrastive (Erraqabi et al., 2021) and Eigenoption (Machado et al., 2018b). Contrastive is representative of a classic approach to representation learning in RL, namely to use a contrastive loss to encode environment dynamics in a latent space, and then guide agent exploration using distance to the latent goal as an intrinsic reward. Figure 5(a) shows an example of such a learned latent space. Eigenoption is representative of how state of the art approaches use the SR, namely by taking the spectral decomposition of the SR (i.e., its eigenvectors) as a basis in which to represent arbitrary value functions. Specifically, Eigenop-

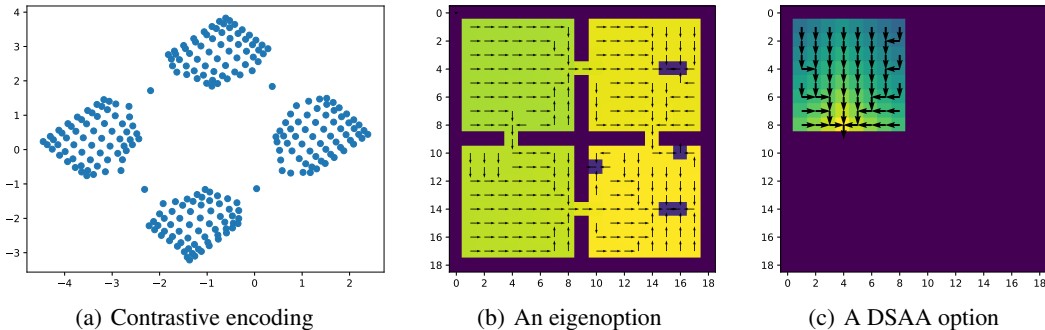

(a) Contrastive encoding     (b) An eigenoption     (c) A DSAA option

Figure 5: On the left we visualize a 2-dimensional latent space learned through contrastive encoding where each point corresponds to one state from FourRooms. In the center we show an example eigenoption, see Appendix A.5 for more examples. Finally, on the right we visualize an example of a DSAA option. Notice our options are restricted to the preimage of a single abstract state (this makes them easier to train than eigenoptions). Lighter color indicates states where the option policies achieve higher value.

tions uses each eigenvector of the SR as an individual reward for an option. See Figure 5(b) for an example of an eigenoption. Finally, Figure 5(c) shows an example of an option output by DSAA.

We test on the FourRooms environment, where for all three algorithms we train using an initial random exploration phase without environment provided reward, then transfer to new tasks in the same environment. We highlight that the sparse nature of reaching a specific state in the environment and the existence of bottleneck states in FourRooms makes this is a relatively difficult task for standard RL. Figure 6 shows the average return of the three methods and Table 1 demonstrates that DSAA is faster and more consistent in reaching a randomly chosen state in the environment. Eigenoptions suffers by not having an abstract state representation to drive exploration towards the goal. Contrastive is slowed down relative to DSAA because it trains an underlying policy over the entire state space, and its shaped reward sometimes leads the agent into dead ends.

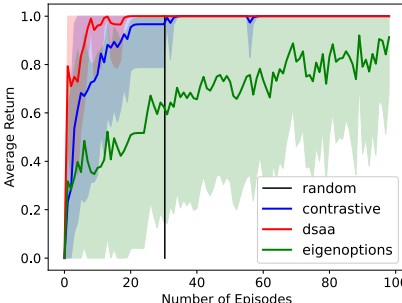

| Method | Mean | Std |
|---|---|---|
| Random | 30.30 | 45.94 |
| Eigenoptions | 8.66 | 10.61 |
| Contrastive | 5.37 | 4.36 |
| DSAA | 1.48 | 0.77 |

Figure 6: We report the average return each method achieves across episodes in the Four-Rooms environment. Results are averaged over 30 randomly chosen starts and goals with episodes of length 200 steps each. The vertical line is the first time random exploration finds the goal.

Table 1: We report the first time each method finds the goal (i.e., diffusion time (Machado et al., 2018b)). This captures the minimum number of episodes before one can consistently solve the task, thus evaluating exploration efficiency while controlling for the effects of training the underlying agent.

## 6.3 ARM2D: MAXIMIZING REWARDS IN A CONTINUOUS SPACE

In these final set of experiments we show the capability of DSAA to maximize rewards online in a continuous control environment. Arm2D is a continuous control task for a three joint manipulator (robot arm) on a 2D-plane (see Figure 7(a)). The arm must move an object (ball) down below a certain height, despite the arm starting below the ball, at which point only it obtains a non-zero

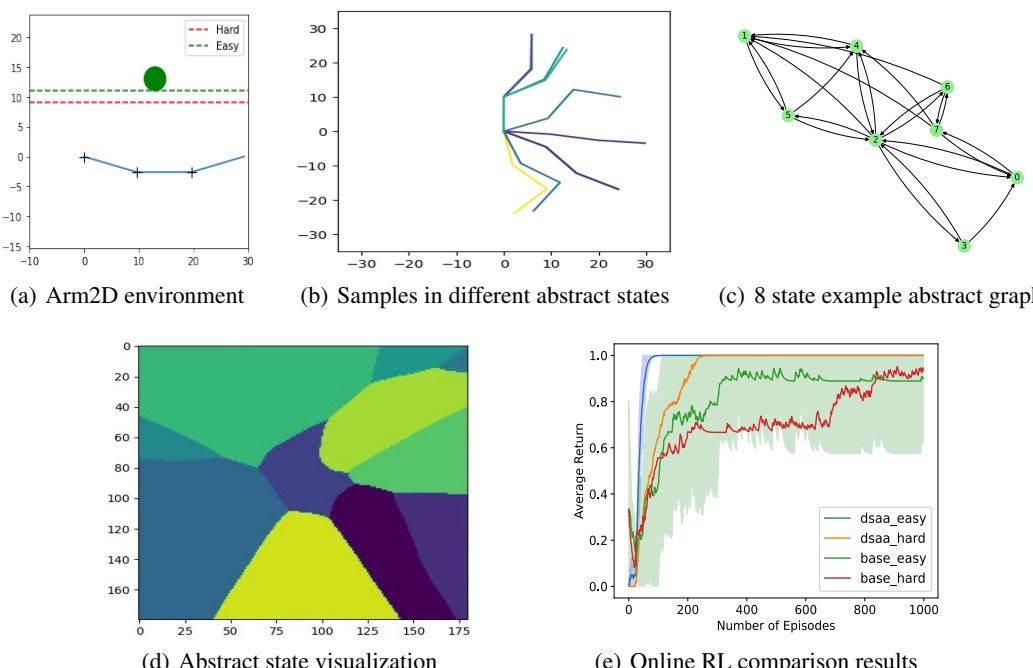

(a) Arm2D environment    (b) Samples in different abstract states    (c) 8 state example abstract graph

(d) Abstract state visualization    (e) Online RL comparison results

Figure 7: Arm2D environment experiments. In (b) we visualize arm positions at boundaries between abstract states; and in (d) we show a 2D slice of the 5D state, colored by abstract state. In (c) we visualize an example abstract graph for 8 abstract states. Finally in (e) we show return curves for DSAA and a Soft Q-Learning baseline on both the "hard" and "easy" tasks shown in (a), averaged over 10 random seeds. The standard deviations, which we only visualize for the "easy" task, demonstrate the consistency of our method.

reward: the lower the height the ball must reach, the more difficult the task, since the sparse reward is more distant. The state space is 5-dimensional, including the three joint angles and the 2D object position. For a more detailed description of the Arm2D environment, we refer to Appendix A.6.1.

The results in Figure 7 demonstrate that we can simultaneously learn an abstraction and maximize reward with few samples. Moreover, the iterative version of DSAA, in which the output options are reused to train a new abstraction, makes exploration progress. In Appendix A.7 we show examples of abstractions for more complex versions of this environment.

## 7 CONCLUSIONS AND FUTURE WORK

We presented a new method to perform a discrete abstraction on the state and action spaces of a reinforcement learning problem using a dataset of transitions in the environment. Our model is based on the successor representation, grouping states together if the agent's behavior after visiting those states is similar. We showed that despite past concerns that gradients from the successor representation would drive any abstraction towards a trivial fixed point, with simple max entropy regularization we can learn a useful abstraction in the discrete setting, and do so in an end-to-end manner. Finally, we showed that a DSAA agent equipped with our action abstraction can solve a variety of downstream tasks.

A limitation of our work is that while our method may help an agent explore the environment efficiently, that exploration is still heavily biased towards the region that has already been explored. Given this limitation, one direction for improvement is to train a higher level policy to bias the walk over the abstract graph towards states which have been identified as the frontier (e.g., see Sekar et al. (2020)). Another interesting line of future work would provide theoretical guarantees on the convergence of clustering by successor representation, as well as other properties related to our method. Finally, we recognize that despite their advantages, discrete representations can struggle to capture very large spaces, and so combining our discrete abstraction with a continuous one could be fruitful.

## 8 REPRODUCIBILITY STATEMENT

All code for running our experiments will be made publicly available along with the final version of this manuscript. In particular, we will make available the reinforcement learning environments on which we test, the random seeds used for generating different experimental runs, and the implementation code and configuration files for training our successor model and DSAA algorithm.

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

## A APPENDIX

### A.1 REVISIONS - COMPARING TO RAMESH ET AL. (2019)

We add a comparison to SuccessorOptions (Ramesh et al., 2019). Figure 8 shows the updated comparison plot and table, Figure 9 shows a comparison of the state partitioning produced by SuccessorOptions, as compared to DSAA. Figure 10 shows examples of the options learned by SuccessorOptions when using four clusters in FourRooms.

We emphasize that SuccessorOptions is implemented on the discrete MDP for which it computes a discrete SR, whereas our method is only given transitions in the environment using the continuous (x,y) coordinate of the state. Thus we simultaneously learn the discrete abstraction and the corresponding SR. This means that while SuccessorOptions must separately cluster states after computing the SR, we find the clustering end-to-end. Moreover, even though our state abstractions are comparable, our action abstraction performs significantly better since, while they navigate to individual primitive states in the environment, we train options that transition between *regions* of the space.

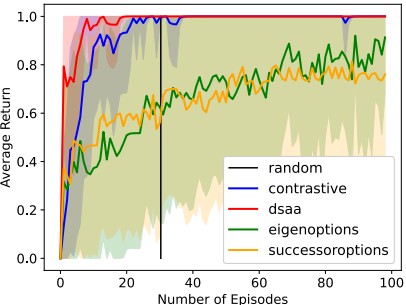

| Method | Mean | Std |
|---|---|---|
| Random | 30.30 | 45.94 |
| SuccessorOptions | 16.30 | 24.37 |
| Eigenoptions | 8.66 | 10.61 |
| Contrastive | 5.37 | 4.36 |
| DSAA | 1.48 | 0.77 |

Figure 8: This new returns chart will replace Figure 6. As expected, Successor Options (Ramesh et al., 2019) performs similarly to Eigenoptions, namely because both learn options which navigate to specific primitive states.

Table 2: This table will replace Table 1. Once again we emphasize the consistency of our method in finding the sparse reward. This consistency comes from the combination of state and action abstraction.

### A.2 REVISIONS - LEARNING AN ABSTRACTION FOR MONTEZUMA'S REVENGE

We train an abstraction for Montezuma's Revenge, the notoriously hardest problem in the Atari suite (Arcade Learning Environment Machado et al. (2018a)). We show results of training a 16 state abstraction in Figure 12 based on random exploration data. Notice that our model $M_{\phi,\psi}$ is capable of learning an abstraction that captures intuitively relevant aspects of the problem, such as the existence of ladders to climb. Moreover notice that relative to the results seen in Figure 5 of Eigenoptions (Machado et al., 2018b), we can use a small number of abstract states and still learn something useful, rather than learning hundreds of options which then would require significant pruning for real applications. While these results don't immediately show the usefulness of our state *and* action abstraction in solving Montezuma's Revenge, they demonstrate that $M_{\phi,\psi}$ generalizes to significantly more complicated domains. Solving the full problem would require coupling our approach with a proper exploration method which would produce the necessary dataset for learning the abstraction; a subject of future work.

### A.3 EXPLAINING MAX ENTROPY

We would like to solve the following optimization problem: given a state space $\mathcal{X}$ compute an abstraction function $\phi : \mathcal{X} \to [N]$, and function $\psi : [N] \to \mathbb{R}^N$, which minimizes the SR temporal difference error (Eq. 2), subject to an entropy constraint:

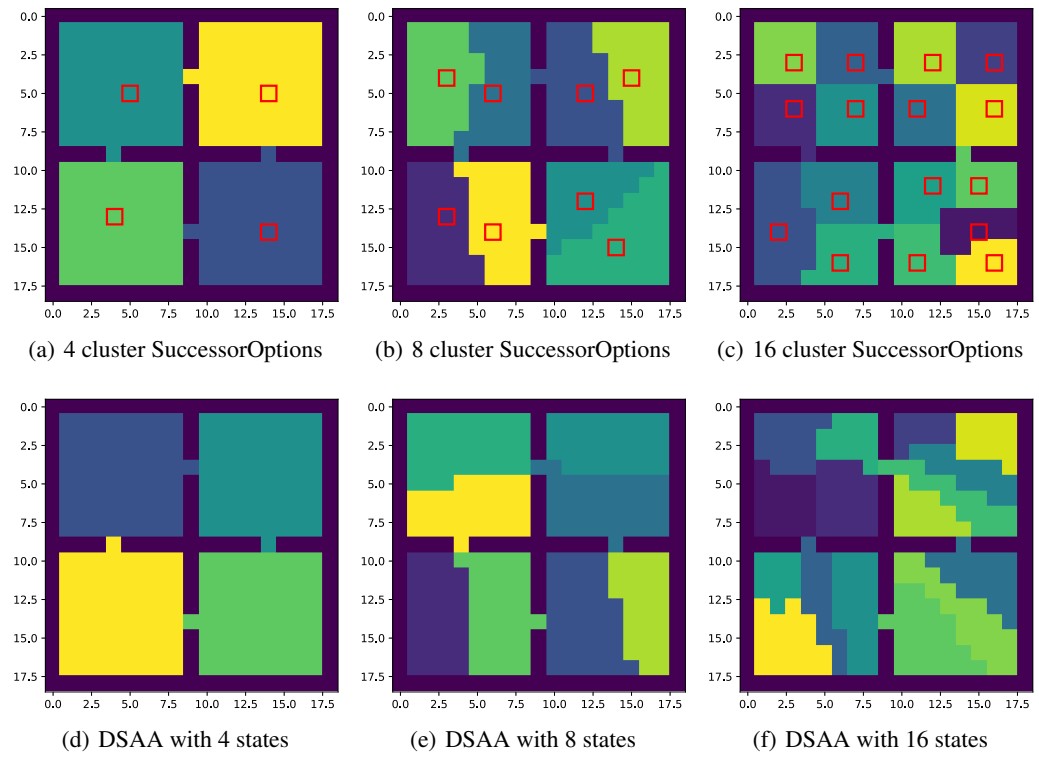

Figure 9: As expected, DSAA and SuccessorOptions partition the state space similarly. Cluster centers for SuccessorOptions are highlighted in red (our approach does not explicitly define a center).

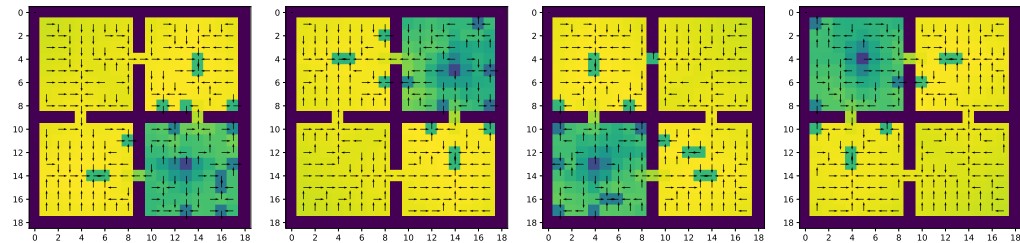

Figure 10: SuccessorOptions in FourRooms with 4 clusters and the corresponding 4 options

$$\min_{\phi,\psi} \quad \mathbb{E}_{x \sim \mathcal{X}, x' \sim p(\cdot|x)} \left\| \psi(\phi(x)) - (\phi(x) + \gamma\psi(\phi(x'))) \right\|_2^2$$
$$\text{s.t.} \quad H(\phi_{\mathcal{X}}) > k \tag{3}$$

where $p(\cdot \mid x) = \mathbb{E}_a p(\cdot \mid x, a)$ and $\phi_{\mathcal{X}}$ is the random variable over $[N]$, which assigns probability $p_{\phi_{\mathcal{X}}}(i) = \mathbb{E}_x(\mathbf{1}_{\phi(\mathrm{x})=\mathrm{i}})$ based on the proportion of $\mathcal{X}$ that gets mapped to $i$ by $\phi$.

We now use the method of Lagrange multipliers and optimize over a batch with stochastic gradient descent to derive the loss $\mathcal{L}_A$ in Algorithm 1. Experimentally we find the method is sensitive to batch size, if the batch size is too small we fail to properly capture the entropy constraint.

## A.4 ADDITIONAL ABLATIONS FOR THE FOURROOMS ENVIRONMENT

In the set of ablations shown in Figure 13, we qualitatively demonstrate the importance of stochastic categorical reparameterization via Gumbel-Softmax as opposed to straightforward classification via Softmax, applied to the output logits of the abstraction model $\phi$. We also explore the effects of

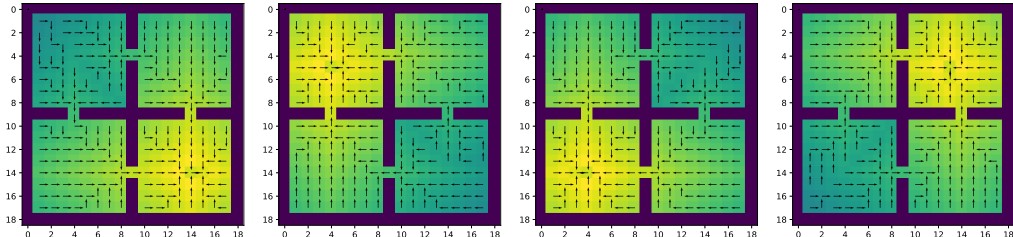

Figure 11: We additionally train options to directly go to the cluster centers for each SuccessorOption. In Ramesh et al. (2019) this is the option they motivate (in prose), though the reward they provide mathematically (shown in Figure 10) is an approximation of this goal. Surprisingly these "desired" options perform worse, and so we display them here merely for completeness.

adding a weight coefficient $\beta > 0$ to the entropy term in the abstraction update loss as $\mathcal{L}_A = \mathcal{L}_H + \beta \mathcal{L}_{SR}$.

In Figure 14 we show that swapping our decoder with a classic reconstruction module and then clustering does not yield good abstractions.

### A.5 Related work implementations

We implemented the comparison methods "generously": for Eigenoption we used the analytical Successor Representation based on the graph Laplacian in the discrete FourRooms environment rather than an empirical deep SR. We then compute eigenoptions and their negatives shown in Figure 15. For Contrastive we note that the original paper (Erraqabi et al., 2021) only reports "acyclic" environments, whereas FourRooms contains multiple distinct homotopy classes of trajectories. We find that the method is harder to tune in such an environment, demonstrating the known result that discrete metrics (of which a simple example is a cycle of 4 nodes with edge length 1) cannot be perfectly embedded in euclidean space (Bourgain, 1985). Their method therefore struggles with certain tasks because the shaped reward (which acts as a heuristic to bias exploration) sometimes leads the agent into dead ends.

### A.6 Arm2D environment details

#### A.6.1 Description

We provide more detail about the Arm2D environment of Section 6.3. The action space consists of 6 discrete actions: each one of the 3 arm joints has two actions which increment or decrement its angle by $\delta$ degrees. For our specific simulations we set $\delta = 1$ degree. Each joint is limited between $-90$ and 90 degrees. The state spaces consists of 5 features: the 3 joint angles and the (x,y) coordinate of the ball. While the joint angles are discretized (by $\delta$), the ball location remains continuous. Each episode was limited to 5000 steps at which point we reset the environment by bringing the arm back to joint angles $[0, 0, 0]$ (horizontal position of the arm), and the ball to $(x, y) = (13, 13)$. The fixed end of the robot arm is located at position $(0, 0)$. The environment reward is a sparse 0 or 1, where 1 is only given to the agent if it moves the ball below the green line ("easy" task at height $y = 11$) or red line ("hard" task at height $y = 9$) shown in Figure 7(a).

#### A.6.2 Model architectures

Our algorithm trains three models: $\phi, \psi, \pi_{s,s'}$. We structure all three as simple feed-forward neural networks. We report the specific values used for the experiments in Section 6.3

- $\phi : \mathcal{X} \rightarrow \Delta(N)$ has two hidden layers with 128 and 256 neurons, and LeakyReLU activation functions. The input is the 5 dimensional primitive state $x$, and the output has $N$ neurons, one for each abstract state, which is passed through a Gumbel-Softmax activation with temperature parameter $\tau$. For these experiments we used $N = 8, \tau = 0.5$.

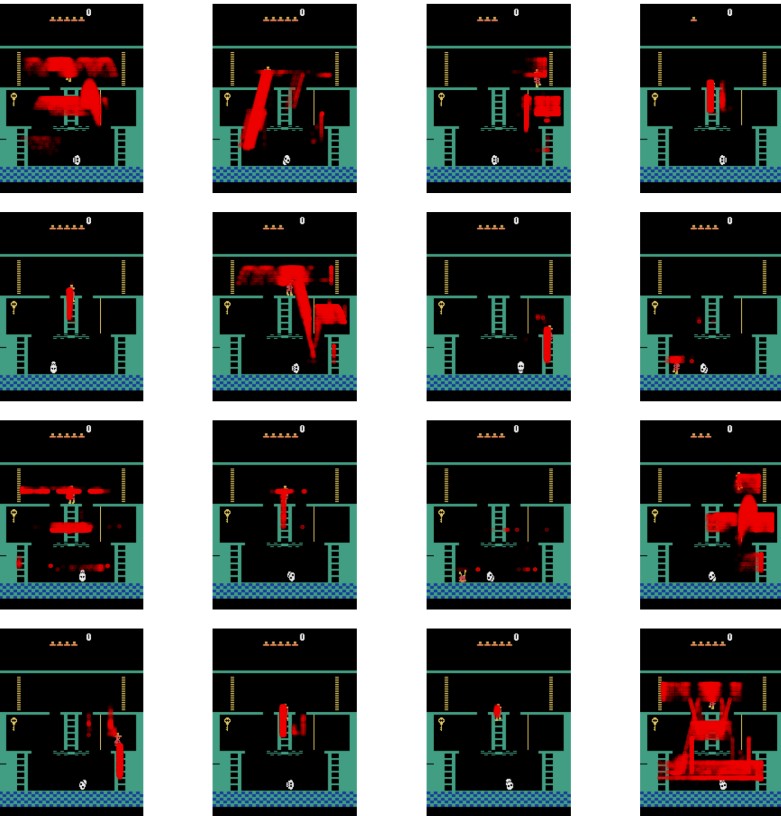

Figure 12: For each of 16 abstract states we visualize samples of of the character's location in the environment in red. The abstraction is trained after exploring the environment using uniform random actions for 200,000 steps. The encoder-decoder model architecture used is the same as in other experiments reported in the paper, with similar hyper-parameter settings. This qualitative evaluation primarily demonstrates that despite the complexity of the environment our model (and corresponding loss) is capable of learning an interesting environment partition. In particular, even without a reconstruction error regularization term in our loss we do not collapse to the trivial abstraction.

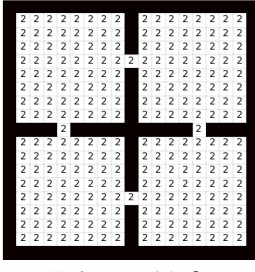 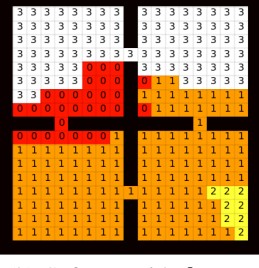 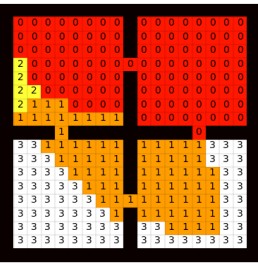

(a) Softmax with $\beta = 1$      (b) Softmax with $\beta = 10$      (c) Softmax with $\beta = 100$

Figure 13: In these ablation experiments we train an abstraction with $4$ abstract states and uniform random exploration. As demonstrated in Figure 4, each abstract state should intuitively correspond to one of the rooms. Replacing Gumbel-Softmax with Softmax leads to poor abstractions: we observe that abstract states occupy more than one room, including the bottleneck connections between rooms, or are even disjoint across two rooms (e.g., abstract state 0 in (b)), which did not occur with Gumbel-Softmax. Moreover, we observe that the size of each abstract state remains unbalanced even with large weight coefficient $\beta$ for the entropy term in the loss.

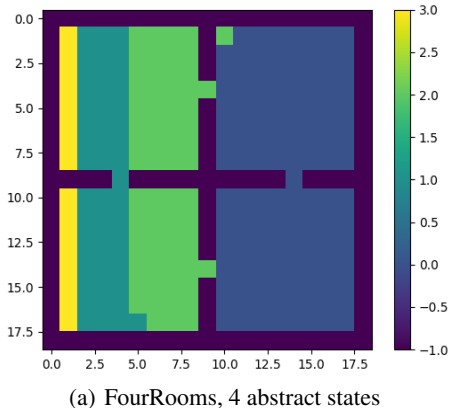 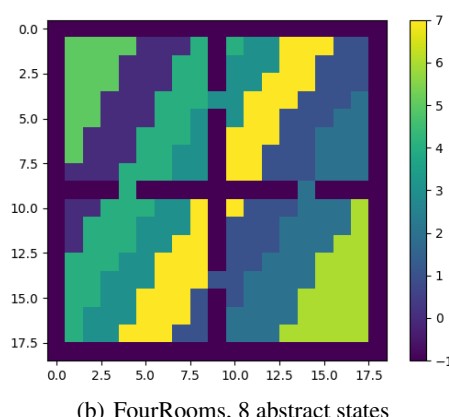

(a) FourRooms, 4 abstract states      (b) FourRooms, 8 abstract states

Figure 14: Using a D-VAE (with a reconstruction loss instead of our SR TD error) generates abstractions that do not take into account agent dynamics and are therefore poor for planning.

- $\psi : \Delta(N) \to \mathbb{R}^N$ has two hidden layers with 64 and 128 neurons and LeakyReLU activation functions. It takes as input the output from $\phi$ and maps it to $N$ neurons output.

- Finally, we group our option policies as a single model with shared parameters except for the final output layer. More precisely, we train a neural network with four hidden layers of 64, 128, 256, 512 neurons and LeakyReLU activation. It takes as input $x$ concatenated with $s = \phi(x)$, which in our case of $N = 8$ abstract states with state vector of dimension 5, results in $8 + 5 = 13$ input states. Then for each option $\pi_{s,s'}$ we have a single linear layer with no activation that maps from the 512 neuron embedding to the discrete actions of our agent, which in our case of $N = 8$ abstract states and $|\mathcal{A}| = 6$ discrete actions, results in $8 \times 6 = 48$ outputs. We note there are other ways we can model the option policies, for example by not sharing parameters between each option policy.

### A.6.3 HYPER-PARAMETER DETAILS

We now describe the hyperparameter settings we used in the experiments shown in Figure 7(e). We highlight that we did not conduct a large scale parameter search for the optimal configuration of the hyperparameters. In this sense our results demonstrate our algorithm without excessive tuning.

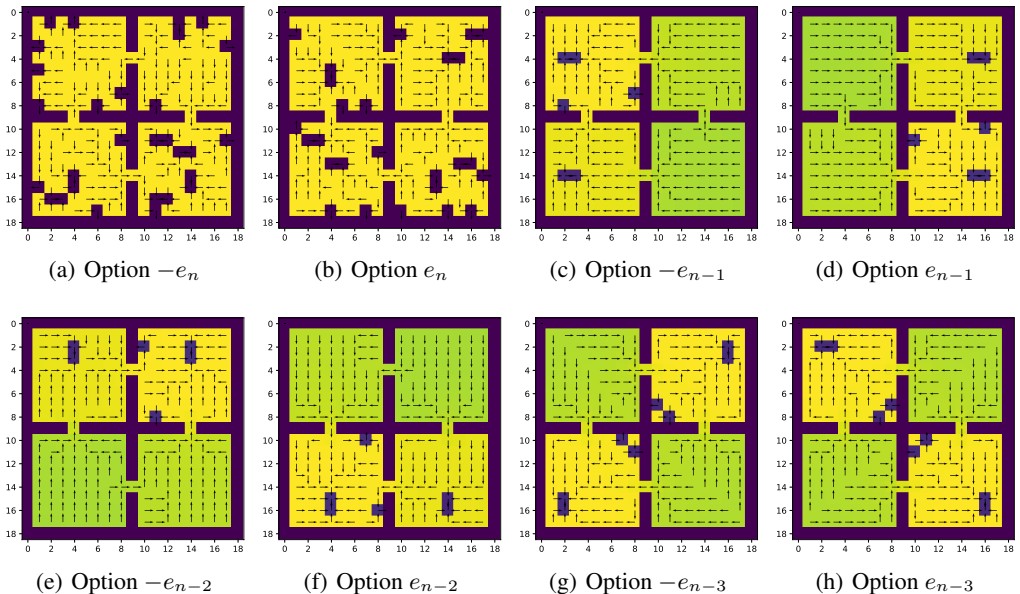

| (a) Option $-e_n$ | (b) Option $e_n$ | (c) Option $-e_{n-1}$ | (d) Option $e_{n-1}$ |

| (e) Option $-e_{n-2}$ | (f) Option $e_{n-2}$ | (g) Option $-e_{n-3}$ | (h) Option $e_{n-3}$ |

Figure 15: The first four eigenoptions and their negatives. Terminal states are shown in black, and regions of higher value are colored lighter (yellow as opposed to green).

We roll out each episode for $5000$ steps and set the replay buffer size for the option policy to $100,000$ transitions (20 episodes). This is the same replay buffer which we use to train the abstraction, and consequently we set the number of exploration steps (E_iters in Algorithm 1) to $100,000$ as well. We use Adam optimizer with learning rate $0.001$ and batch size $512$ for updating both $\pi_{s,s'}$ and $\phi, \psi$ (which are updated in tandem as an encoder-decoder pair). Both the update equations for Q-learning and the successor representation have a discount factor $\gamma$, which we set to $0.95$. We use a target Q model for computing the temporal difference target, which updates with a delay behind the online model of 20 iterations. We reward each option policy when it transitions to the correct next abstract state with 200 reward points – this reward needs to be high enough to balance the entropy based reward the soft policy receives on each transition, but is otherwise arbitrary.

We implement our models using PyTorch, without any external learning libraries or outside implementations of Soft Q-learning. We run on CPU using an Intel(R) Core(TM) i9-10980XE CPU @ 3.00GHz. The code can be found as part of the Supplementary Material.

## A.7 SCALABILITY OF $M_{\phi,\psi}$

As mentioned in the final sentence of Section 7, discretization is not always the best approach for particularly large spaces in which one may want to combine discrete and continuous representations. Nevertheless, in this section we demonstrate that our method for abstraction scales to both image representations and high dimensional agents. Figure 16 demonstrates that when using a more complex input representation, such as an image, our model is still capable of learning a reasonable abstraction consistent with prior results. Figure 17 shows a learned abstraction for a 10 joint robot arm in the plane, with one joint fixed at a point and all others allowed to vary between $-\pi$ and $\pi$. Note that self collisions are not valid configurations.

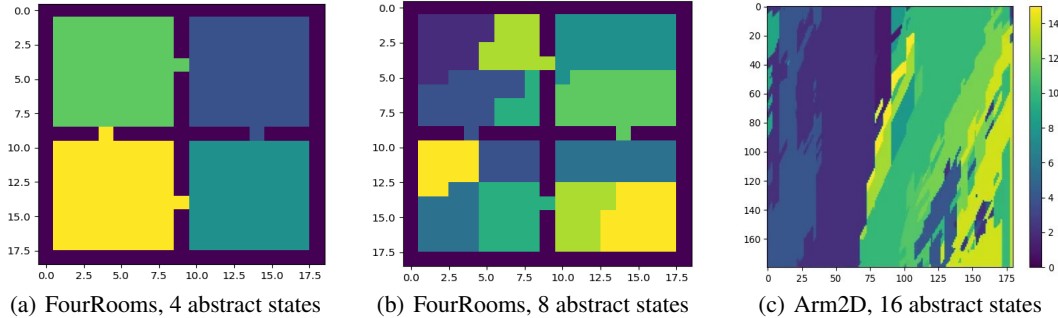

(a) FourRooms, 4 abstract states    (b) FourRooms, 8 abstract states    (c) Arm2D, 16 abstract states

Figure 16: Using the same environments reported in the main body of the paper, we demonstrate that replacing the input representation with images still yields qualitatively reasonable (and similar to previous) discretizations of the space. In FourRooms the input is $19 \times 19 = 361$ pixels, with a single pixel sparsely marking the agent location. In Arm2D the joint angles are replaced by $62 \times 62 = 3844$ pixels depicting the arm. Note in Arm2D the resulting discretization is much less smooth relative to Figure 7(d).

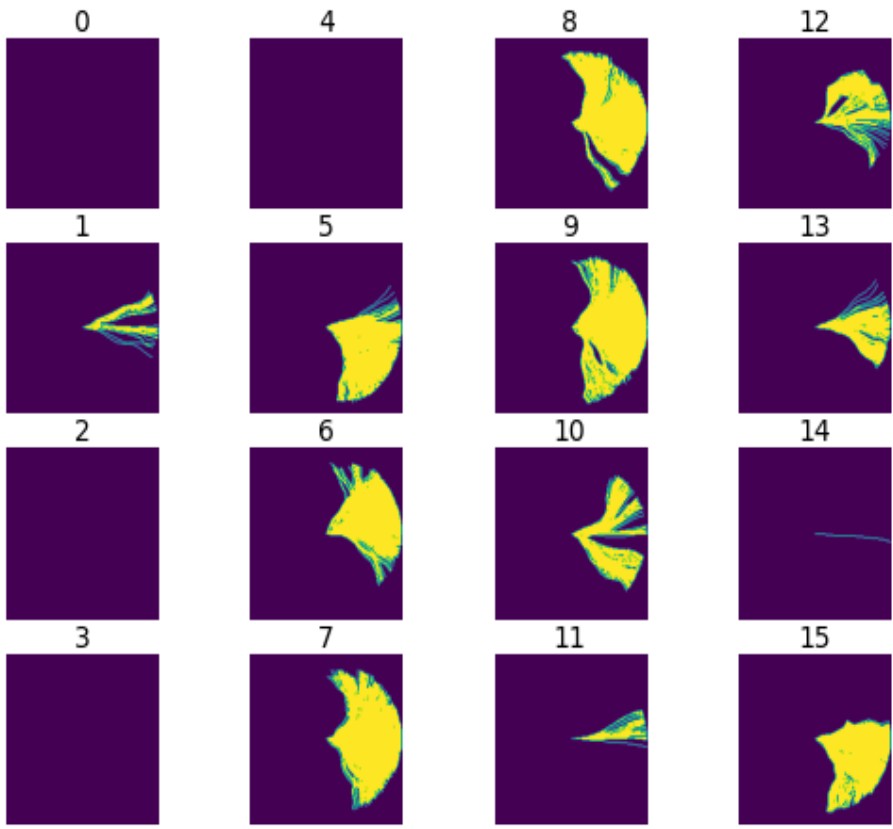

Figure 17: For each abstract state we visualize samples of configurations of a 10 joint arm. Notice that some states, such as 0 and 4, have no arm configurations. One could increase the max entropy coefficient term to encourage the model to more evenly distribute the samples. There is no obvious intuitive partitioning of such a configuration space in the absence of narrow passages which would be formed by obstacles.

