# OpenReview forum: "Discrete State-Action Abstraction via the Successor Representation"
_ICLR.cc/2023/Conference — Submitted to ICLR 2023_

### Official Review · Reviewer_TM9s · 2022-10-24

**Confidence:** 3
**Correctness:** 3
**Technical Novelty And Significance:** 3
**Empirical Novelty And Significance:** 1
**Recommendation:** 3

**Clarity, Quality, Novelty And Reproducibility:**

Quality: The quality of the writing and plotting is OK.
Clarity: The writing is somewhat vague on the motivation and the methodology is a bit underspecified (potentially because of the complexity of the method).
Reproducibility: The authors claim the paper will be released to the public but there's no code in the current submission. Since the whole algorithm seems quite complex, I think it may not be very easy to reproduce the results.

**Strength And Weaknesses:**

Strength:
1. The idea of learning a graph and using it for hierarchical decision-making is interesting on a high level.

Weaknesses:
1. Environment of all the tasks is of very low degree of freedom (Dof=2), and the optimal policy is very simple. It's unclear whether the method can be applied to tasks with practical value.
2. The motivation of a lot of the novel designs (the successor representation loss in eq (2), lazy random walk for reward maximization) is not well-justified and seems like can cause problems in a general setting.
3. The baselines this method is comparing to in section 6.2 are within their framework. It's more like ablation of representation learning component of DSAA. No proper offline/online, model-based and model-free RL methods are compared to the proposed method.

**Summary Of The Paper:**

This paper proposes to learn a successor representation and use it to cluster the states. The clusters are then treated as nodes of a graph and an (option) policy is trained to navigate the graph. The graph can be used in either a goal-oriented setting by searching; or a reward maximization setting with a lazy random walk.

**Summary Of The Review:**

This paper is an empirical one where a lot of design choices are based on the authors' intuition. However, the paper didn't provide strong empirical evidence that the method they proposed actually works in tasks with practical value. All the tasks are quite simple (2d navigation and 2d arm) and working on them is more of a proof of concept. There could be so many implicit hypotheses that the method relies on if they can only work on tasks with this low degree of freedom. Even for these toy tasks, it's not clear whether the method has any advantages over well-established control/RL methods due to the lack of baselines.
I think this paper is not ready for acceptance for ICLR.

---

> ### Author Response · Authors · 2022-11-28
> **Responding to individual concerns**
>
> We are responding to the weaknesses you mention regarding our work in-line below:
>
> - Environment of all the tasks is of very low degree of freedom (Dof=2)?
>   - Please note that our arm has 3 DoF, and moreover we include the location of the ball in the state which makes it a 5 dimensional search space. This is low but not very low, and we do display the state abstractions for higher DoF in the Appendix.
> - The motivation of a lot of the novel designs (the successor representation loss in eq (2), lazy random walk for reward maximization) is not well-justified and seems like can cause problems in a general setting.
>   - Our loss is very well justified in many prior works. It is further justified in Section A.3 of the Appendix.
>   - We would like to address this concern, but it would be helpful if you share what problems you foresee, and what are the characteristics of this “general setting” you have in mind?
> - The baselines this method is comparing to in section 6.2 are within their framework. It's more like ablation of representation learning component of DSAA. No proper offline/online, model-based and model-free RL methods are compared to the proposed method.
>   - We are not sure we understand this criticism. The representation learning component of DSAA (meaning M_{\phi, \psi}) is our main contribution. Naturally we perform ablations on it.
>   - As we’ve stated in other responses, this is an abstraction work and as such we compare to abstraction literature (at this point *three* other methods). Nevertheless, we do not understand the characterization of these related works as “not proper RL”.
> - You indicated that the method seems complicated, and code was not provided
>   - We have provided code and pointers for where to look in the code to find our *simple* implementation.

---

### Official Review · Reviewer_C2ny · 2022-10-26

**Confidence:** 4
**Correctness:** 4
**Technical Novelty And Significance:** 4
**Empirical Novelty And Significance:** 4
**Recommendation:** 8

**Clarity, Quality, Novelty And Reproducibility:**

The paper is clearly written and the work is novel as far as I am aware.
Because the method is relatively simple and cleanly described it should be reproducible. The authors state that the code will be made available upon acceptance.

**Strength And Weaknesses:**

This is a well written paper which clearly describes its motivations, contributions as well as limitations. The method is theoretically grounded and seems sound. The approach is novel as far as I am aware and cleanly described. The experiments are well thought-out and appropriate comparisons are made to prior approaches.
The environments are a bit simple, but given that prior approaches are less effective/noisy on them, that is justified.
More quantitative results could have been included for different exploration policies for learning the success features/abstractions.

**Summary Of The Paper:**

The paper outlines an approach to use successor representations to drive a discrete state space abstraction. The abstract states are clusters in successor space, so "nearby" states have similar successors. The paper also contributes a way to utilize the abstraction learned to solve tasks efficiently by interpreting transitioning between abstract states as options, and outlines an algorithm to jointly learn the successor representations, the abstract representation and the option policies.

**Summary Of The Review:**

This is a good quality paper, cleanly written and justified with appropriate experimentation and I recommend acceptance to ICLR.

---

### Official Review · Reviewer_icxB · 2022-10-29

**Confidence:** 4
**Correctness:** 3
**Technical Novelty And Significance:** 2
**Empirical Novelty And Significance:** 2
**Recommendation:** 3

**Clarity, Quality, Novelty And Reproducibility:**

The main novelty in the paper is the specific way the successor representation is used in defining the abstract states. An earlier paper by Ramesh et al. (IJCAI 2019) has used the same intuition of grouping states together using the successor representation. Generally, the central ideas and components of the approach have appeared earlier in the literature but the proposed approach puts them together in a novel way.

The paper is well organised but the individual sections could be better written, with more clarity and detail.

I have not spotted any errors. But I would note that the experimental analysis is not strong enough to back the claims of the authors regarding how useful the approach is.

**Strength And Weaknesses:**

Strengths:

-- The authors present a plausible approach to abstraction in reinforcement learning, building on several earlier ideas and methods.

Weaknesses:

-- The analysis and evaluation of the approach is quite limited.

First, the authors explore the behavior of the proposed approach in a relatively narrow range of environments. The first domain is a small gridworld with four rooms. The authors state "We highlight that the sparse nature of reaching a specific state in the environment and the existence of bottleneck states in FourRooms makes this is a relatively difficult task for standard RL." Not many people would agree with this statement. On the contrary, this is quite a simple task for reinforcement learning. This domain is not only easy but also has a very simple structure, and many alternative methods would identify very similar abstractions (e.g., those based on graph clustering). It is not a particularly informative environment. The second domain is more complex than the gridworld but still relatively simple.

Secondly, the analysis in each domain is relatively limited and does not give the reader a good understanding of the behaviour of the proposed approach. For instance, there is no exploration of how agent performance varies with the number of abstract states.

Thirdly, the analysis does not explore some existing approaches that are closely related. For example, as the authors note, Ramesh et al. (IJCAI 2019) propose a similar abstraction of the environment using the successor representation, which would be an informative baseline. Approaches based on graph cuts/clustering would also be relevant and informative. Some of these methods may not scale as well as the proposed approach but it would still be useful to see their similarities/differences and relative strengths/weaknesses explored.

-- How the approach would fare in stochastic environments is not discussed or explored.

-- Computational complexity of the approach is not discussed or explored.


**Summary Of The Paper:**

The authors propose an approach to learn a discrete abstraction of an environment in reinforcement learning. The proposed approach creates abstract states using the successor representation (Dayan, 1993), reflecting the intuition that states should be grouped together based on "what happens after visiting a state", and creates abstract actions (options) that take the agent from one abstract state to neighbouring abstract states. The authors present an empirical analysis of the proposed approach in two domains: a gridworld with four rooms and a three-joint control task on a two-dimensional plane.


**Summary Of The Review:**

The authors present a plausible approach to abstraction in reinforcement learning, building on several earlier ideas and methods. The main novelty in the paper is the specific way the successor representation is used in defining the abstract states. The approach could prove useful but the analysis in the current paper does not give the reader a good understanding of the behaviour of the approach and its strengths/weaknesses relative to existing methods.

---

> ### Author Response · Authors · 2022-11-28
> **What is "easy" for RL?**
>
> We wish to respond to your first comment regarding the difficulty of FourRooms and our claim that standard RL struggles in this environment.
>
> First we want to clarify our language since it may not have been entirely clear - by "standard RL" we refer to reinforcement learning without abstraction or intrinsic motivation, e.g., a standard model free algorithm (we highlight that such algorithms are still SOTA and the focus of much research).
>
> FourRooms has a total of 260 discrete states, that is very small! Yet random exploration takes tens of thousands of steps (in the model-free scenario) to explore certain states. With a capped episode length of 200 steps random exploration takes *hundreds of thousands* of steps to fully explore the entire environment. This is what we mean by difficult - that relative to the size of the environment, in the absence of abstraction, the exploration phase is significantly longer than it intuitively needs to be. We admit our language should have been more clear.
>
> We did not come up with this environment, it has been used for decades as a prime example of the need for abstractions (both state and action). We also note that our environment is significantly larger than the FourRoom environment in Machado et al. (2018) which contains 104 states.
>
> Finally, we disagree that this is not an informative environment. Even though past methods may identify similar abstractions, they do not do so end to end, and we demonstrate that our specific combination of state and action abstraction outperforms related methods. Testing our model on the same environment many others have tested their methods on is quite valuable. Moreover, this environment is perfect for verifying intuition against empirical results, where it's important to note that some methods, such as Bisimulation, find quite different abstractions that don't match this intuition.
>
> Machado, M. C., Rosenbaum, C., Guo, X., Liu, M., Tesauro, G., & Campbell, M. (2018). Eigenoption discovery through the deep successor representation. **ICLR 2018**.

---

### Official Review · Reviewer_vZz4 · 2022-11-04

**Confidence:** 4
**Clarity, Quality, Novelty And Reproducibility:** The paper is written well with clarit…
**Correctness:** 3
**Technical Novelty And Significance:** 2
**Empirical Novelty And Significance:** 2
**Recommendation:** 5

**Strength And Weaknesses:**

## Strengths
- The paper is clearly written, easy to follow, has a good structure and flow, and contains sufficient background to understand the method.
- The idea of clustering states into a discrete set of clusters and using them for option learning is interesting and seems novel.
- The method is simple, intuitive, and achieves the paper's desiderata of uniformly sized state clusters with similar Successor Representations.

## Weaknesses
- **Are the assumptions practical?**
    + My main concern is that the assumptions required to make the method work do not seem to be practical and would not scale to complex tasks. For instance,
        * The idea of defining state clusters only makes sense in small and closed environments. In unknown and complex environments, it is not even clear what the number of state clusters (and options) would be.
        * The dataset of environment transitions should cover all the states in the environment, at least those that the downstream task would observe.
        * In search-like tasks like mazes or Arm2D, the main challenge is reaching certain states using a reward. But this paper bypasses this problem of state exploration by assuming the dataset does most of the hard job. Therefore, it does not seem practical for most problems.
        * Equally-sized state clusters: Depending on the task family, differently-sized clusters might make more sense, e.g., guided by reconstruction or environment reward or reachability of states or semantic properties (such as objects) of states.
    + Are there realistic applications that would still fall under such assumptions and thus benefit from the proposed method?
- **Experimental Evaluation**
    + **Environments**
        * While the proposed method can work in both discrete and continuous settings, I am concerned about the complexity of tasks it can scale to. The current environments are too simplistic and deterministic: even random exploration can find the goal in 30 episodes in the FourRooms environment. The prior work (Machado et al. 2018) also uses Successor Representations and shows results on the Atari domain.
    + **Mismatch from claims**: While the paper claims (in the introduction) that discrete state abstraction is helpful to discover discrete objects or properties and for understandability, none of these benefits are exhibited in the experiments.
    + **Baselines**: The use-case of the proposed method is in learning state abstraction and options unsupervisedly. However, several skill discovery methods, such as Pertsch et al. (2021), discover skills from unsupervised offline datasets and use them to accelerate downstream RL. Shouldn't such Hierarchical RL methods also be compared as baselines to demonstrate the importance of discrete state abstractions?

[1] Marlos C Machado, Clemens Rosenbaum, Xiaoxiao Guo, Miao Liu, Gerald Tesauro, and Murray Campbell. Eigenoption discovery through the deep successor representation. In 6th International Conference on Learning Representations, 2018.

[2] Pertsch, Karl, Youngwoon Lee, and Joseph Lim. "Accelerating reinforcement learning with learned skill priors." Conference on robot learning. PMLR, 2021.


**Summary Of The Paper:**

This paper proposes to use a dataset of environment transitions to divide states into N discrete clusters. They propose a Discrete VAE architecture that encodes a state into one of N discrete states and then decodes it into its successor representation. The encoder is regularized by a uniform prior leading to similarly sized clusters of states and the decoder ensures states with similar Successor Representations have the same discrete cluster. Temporally abstracted actions (options) are trained to transition between the discovered state clusters. This procedure is shown to perform well in two simple environments: FourRooms (discrete) and Arm2D (continuous).

**Summary Of The Review:**

The paper has original ideas and is written well. However, I am not yet convinced of the importance of the problem solved and method proposed, as it may be limited to toy tasks — both, in terms of the assumptions and the experiments. I would consider raising my score if these concerns could be addressed.

---

### Author Response · Authors · 2022-11-09
**Thanks to the reviewers for their comments**

We thank the reviewers for their thoughtful comments, and wish to address some shared concerns in this response.

Your main concerns have focused on our experiments, namely on our choice of comparison methods and the complexity of the environments we experimented on.

**Comparison methods**
- Firstly, as Reviewer *icxB* correctly points out, we should compare to Ramesh et al. (2019). We have now done so and those results are in Appendix A.1.
  - As Reviewer *C2ny* highlights, we improve on their results in two ways:
    - Our state abstraction is more general, and can be learned end to end rather than using a separate clustering stage
    - Our corresponding action abstraction is more consistent (e.g., because their termination sets are individual primitive states)
- We feel that otherwise our current set of baselines do a good job of covering the approaches seen in the literature.
  - While DSAA can be seen as a general purpose RL algorithm, our main contribution (highlighted in section 1.1) is a *representation learning method that uses the SR in a new way*. We took this into account when choosing baselines (explained in more detail in section 6.2).
  - Reviewer *vZz4*, while we appreciate your suggestion, Pertsch et al. (2021) is an action abstraction paper whereas ours focuses on state (and action) abstraction. Moreover, we already compare to one action abstraction (i.e., skill-learning) work through Eigenoptions.

**Complexity of environments**
- We recall that the problem in question is abstraction, in which we assume that the underlying environment is simple but the input representation is not.
  - Our environments may be simple, but our input representations are not.
  - First we emphasize that our method does not use the true discrete graph of FourRooms, but rather the “continuous” (x,y) location.
  - Using continuous representations is one level of complexity, but we also draw your attention to the ablations performed in the Appendix, where the input representation consists of high dimensional images or is augmented with random noise. We apologize if these results were initially hard to find.
- Prior work on abstraction and the successor representation in complex environments has often consisted of *qualitative results*.
  - For example, Reviewer *vZz4* rightly points out that Machado et al. (2018) test their method on the Atari domain, but we note that the results in Figure 5 of Machado et al. (2018) are qualitative, merely showing examples of learned option terminations superimposed on the environment.
  - We already provide similarly qualitative results on a 10 joint version of our arm in Appendix A.5 (A.6 in the new Appendix), and feel that adding more such results on a complex environment such as Atari does not improve the manuscript

**Code**

As promised (and requested by Reviewer *TM9s*), we now provide our code for reproducibility. We have added pointers in the README to direct your attention to the main functions (in *update_models.py*) to demonstrate that the overall algorithm is simple. *transfer_experiments.py* demonstrates that using our method is similar to pre-existing algorithms, and that our experimental results are reproducible.

*Note that for your convenience, all updates to the manuscript are in Appendix A.1, and will naturally be incorporated into the final version as necessary.*

[1] Ramesh, R., Tomar, M., & Ravindran, B. *Successor options: An option discovery framework for reinforcement learning*.  In International Joint Conference on Artificial Intelligence, 2019.

[2] Pertsch, Karl, Youngwoon Lee, and Joseph Lim. *Accelerating reinforcement learning with learned skill priors.* Conference on robot learning. PMLR, 2021.

[3] Marlos C Machado, Clemens Rosenbaum, Xiaoxiao Guo, Miao Liu, Gerald Tesauro, and Murray Campbell. *Eigenoption discovery through the deep successor representation*. In 6th International Conference on Learning Representations, 2018.

---

> ### Author Response · Authors · 2022-11-19
> **Update to manuscript: added Atari experiment**
>
> Dear reviewers,
>
> We've updated the manuscript one last time, namely Section A.2 of the Appendix now contains results on Montezuma's Revenge, an Atari game included in the Arcade Learning Environment (Machado et al., 2018a). We thank the reviewers for the suggestion to test on more complex environments as the results have been interesting, and hope that, while qualitative, they help address some of your concerns regarding the scalability of our method.
>
> [4] Marlos C. Machado, Marc G. Bellemare, Erik Talvitie, Joel Veness, Matthew J. Hausknecht, and Michael Bowling. *Revisiting the arcade learning environment:  Evaluation protocols and openproblems for general agents*. Journal of Artificial Intelligence Research, 61:523–562, 2018a.

---

> ### Comment · Reviewer_vZz4 · 2022-11-23
> **Several concerns left unaddressed**
>
> Thanks for the rebuttal and I apologize for the late response.
>
> I have gone through all other reviews and the author response thoroughly. While I appreciate the additional baseline result and Atari visualization that the authors provide, I think there are several concerns of mine and the other reviewers that were left unaddressed:
> - Are the assumptions practical? (vZz4, icxB, TM9s)
>     + e.g. scaling to more complex environments, working in stochastic environments.
>     + Read my review for detailed concerns.
> - Current set of environments is too simple and deterministic. (vZz4, icxB, TM9s)
>     + The authors say that "Our environments may be simple, but our input representations are not". However, the input representations are still very simple compared to what common Deep RL benchmarks work on. Is there a good justification for why more complex environments are not evaluated?
>     + Atari results are only qualitative. The benefit of state abstraction must be seen in the downstream learning. So, do the learned options really help to solve the Atari task better than conventional RL approaches?
>     + Even still, the problem still remains that this approach is limited to the kind of data that one starts with, as the SR and options are learned in the states already in the dataset. This is why more complex and general environments are needed to justify the benefits of this method. "A representation learning method that uses the SR in a new way" as a contribution is justified if the learned discrete representations are indeed useful in general.
>
>
> - About the new results added in comparison to Ramesh et al. (2019) — the state abstraction does not look any different.
>     + The paper's primary contribution is about *learning a discrete abstraction by partitioning an arbitrary state space from a dataset of transitions which explore that same space*. But, as Figure 9 demonstrates, the state abstraction obtained by SR + clustering (Ramesh et al. 2019) is similar to DSAA (this paper).
>     + The key benefit comes from action abstraction, but if I understand correctly, that can be directly clubbed with Ramesh et al (2019)'s clusters and the performance should be the same as DSAA?
>     + While I appreciate the value of showing that this clustering can be learned end-to-end using Gumbel Softmax, its benefit is not demonstrated (as it seems to come from the action abstraction), at least in the environments shown in this paper.
>
>
> Therefore, as mentioned in my original review, the limited scope of the method in terms of assumptions and experiments needs to be addressed in a future iteration of the work. I will maintain my rating.

---

> > ### Author Response · Authors · 2022-11-28
> > **Responding to specific concerns (part 1)**
> >
> > - Are the assumptions practical?
> >   - Working in stochastic environments?
> >     - First we emphasize that the successor representation implicitly accounts for stochasticity (since it merely depends on the dynamics), and our random exploration dataset already introduces randomness.
> >     - More generally we do not understand the value of environment stochasticity in this context, our main concern is size of the search space (diameter) and bottleneck dynamics.
> > - In your previous review you mentioned three assumptions:
> >   - Discrete clustering?
> >     - We have motivated in depth why discrete abstraction is desirable. We have also shown that it yields more efficient and consistent exploration.
> >     - Working with discrete representations is notoriously difficult - it took lots of experimentation to get it right. This final success is a contribution in itself.
> >     - Yes, discrete representations have drawbacks! Our view is that extending discrete representations to arbitrarily large environments is **beyond the current scope**. One of the main goals of this manuscript is to lay the groundwork for such a method, e.g., one which performs discrete abstraction hierarchically.
> >   - Dependence on a dataset of transitions?
> >     - This seems like a technical misunderstanding based on our phrasing of the contribution. Yes, our method depends on a dataset, but this is true for every other RL and abstraction method. Many if not most simply resort to the baseline of random exploration. We do not provide a better method for generating such a dataset (and do not claim to do so), but only a new method for using such a dataset.
> >       - While we could change our phrasing to simply depend on random exploration, we found the experiments in section 6.1 illuminating (*and lacking in other works*), as they demonstrate how changing the exploration changes the abstraction.
> >     - We **do not** “bypass the challenge of sparse reward by assuming the dataset does the hard work”. The initial dataset contains no reward, then the abstraction is learned, and then used in a standard setting (without prior exploration) to find the reward. This is the classic unsupervised setting, in which one transfers a learned representation to future tasks.
> >   - Equal sized clusters?
> >     - You are right, clusters should not always be the same size. We emphasize that our method uses “same size” as a penalty not a constraint, and as you can see from our results the clusters we learn are not always the same size.
> > - Current set of environments is too simple?
> >   - Input representations are still very simple compared to what common Deep RL benchmarks work on?
> >     - We emphasize that we do have experiments testing with images, which is not less complicated than common DeepRL benchmarks. In Atari we tested with the 128 dimensional RAM state.
> >     - It is misleading to imply that the standard in the literature is to use complex input representations. As members of this community we have often walked away from reading a paper with the belief that they operated on images, only to later find out this was not the case. *Moreover, while we have provided code, most do not*.
> >     - Most importantly, this is not a Deep RL paper. This is an abstraction paper. The kinds of environments which SOTA abstraction methods are capable of representing lags behind SOTA massive scale deep RL. We are not trying to maximize an individual reward in a single environment, we are trying to come up with a good representation that generalizes to many tasks.
> >       - It is worth noting that the amount of exploration we perform is orders of magnitude less than Deep RL works.
> >     - Nevertheless, we would be happy to consider any work that you could refer us to in our setting of abstraction where more complex representations are used.
> >   - Atari results are only qualitative?
> >     - First, we emphasize that the paper which you (and we) cited when referring to “other works which test on atari” presented similar (and less informative) qualitative results. Qualitative results are still results, and in this case they are representative of the literature in abstraction. You requested these experiments, citing a specific work, and we produced complementary results.
> >     - Second, our representation depends on the exploration method used to generate the dataset. Currently, we do not have a good way to explore such complicated environments more efficiently - random exploration fails. Improving such exploration is **beyond the current scope**.
> >     - Finally, we believe that these qualitative results should be considered in the context of our other quantitative ones, and only serve to strengthen them.

---

> > ### Author Response · Authors · 2022-11-28
> > **Responding to specific concerns (part 2)**
> >
> > - About the new results added in comparison to Ramesh et al. (2019) — the state abstraction does not look any different?
> >   - We may be missing something regarding this concern since we do expect similar results to Ramesh et al. (2019). We have highlighted multiple reasons why we improve on their abstraction, namely that we *extend to input spaces that are not discrete*. This is an important and valuable generalization.
> > - The key benefit comes from action abstraction, but if I understand correctly, that can be directly clubbed with Ramesh et al (2019)'s clusters and the performance should be the same as DSAA?
> >   - First, as mentioned above, this is *not the only benefit* (namely we now perform the abstraction end-to-end for more general environments, which is a huge advantage).
> >   - Second, while we use the successor representation (psi) to learn our abstraction, afterwards our abstraction model (phi) stands alone. This is much more conducive to future improvement which generalizes even when the environment dynamics change. In other words, the option reward in Ramesh et al. (2019) explicitly depends on the SR of every state, whereas ours does so implicitly.
> >   - Finally, even without our improvements on the state abstraction, merely proposing a new type of action abstraction is itself a contribution. Moreover we note that our experiments clearly indicate this new action abstraction is in fact better.
> > - The benefit of Gumbel is not demonstrated?
> >   - Again, we would like to understand the source of this concern as we do provide evidence of the benefit of Gumbel, primarily through ablations in the Appendix. Such experiments are naturally not extensive, as there is not much to say when something (baseline softmax) does not work other than “this did not work”… These results are anecdotal and in the Appendix because, from a scientific perspective, we have not proven that baseline softmax will not work, which is not an easy task.
> >     - In those ablations you will notice that the benefits of Gumbel come in the state abstraction, not the action abstraction. In short, Gumbel helps avoid degenerate representations.

---

> ### Comment · Reviewer_TM9s · 2022-11-23
> **Concerns Left and Suggestions**
>
> I'd like to thank the authors' response, code release and experiments.
> My major concern is the empirical evaluations are in a quite limited setting in terms of control degree of freedom, and the argument and qualitative Atari experiments, unfortunately, didn't address my concern.
>
> The authors claim on the contribution is "We describe an algorithm to apply our model, named Discrete State-Action Abstraction (DSAA), which computes an action abstraction in the form of temporally extended actions, i.e., Options, to transition between discrete abstract states."
> I would assume the temporally extended and discrete actions are an important essence of the method. I don't think input representations in the pixel domain are relevant to the action abstraction. Atari experiments are definitely more interesting than the 2-Dof arm but they also have a well-defined discrete action space so the discretization is no longer important. Also as, vZz4, without combining the proposed method with a well-established algorithm on Atari and testing whether it really helps for the performance, it's hard to tell if the method is really helpful.
>
> For the next iteration of the work, I'll suggest the authors test it on continuous control tasks like locomotion control (with higher Dof) or ideally more complex tasks. In these settings, clustering the actions both in action space and temporal space seems to be more sensible.

---

> ### Comment · Reviewer_icxB · 2022-11-24
> **Concerns remain after rebuttal**
>
> Thank you for the rebuttal. I appreciate the additional results and their clear description here and in the revised paper. I will not be able to raise my score because the new results do not (yet) go very far in addressing the concerns raised by multiple reviewers. I highly encourage the authors to expand and deepen their analysis of the proposed approach.

---

> ### Author Response · Authors · 2022-11-28
> **Addressing your remaining concerns**
>
> Dear Reviewers,
>
> Firstly, at a high level, we want to emphasize what this manuscript is not about. **This is not a deep reinforcement learning work**. The algorithm we presented does not solve arbitrary MDPs for two main reasons: discrete abstractions (as you’ve mentioned) cannot partition arbitrarily complex environments, and our algorithm does not include an exploration method beyond the region captured by the provided dataset. Fixing these deficiencies to produce a full RL algorithm is the subject of ongoing work, but is **beyond the scope** of the current one.
>
> This manuscript is about abstraction. Consequently, our experiments test various properties of our abstraction. We examine sensitivity to initial dataset, whether due to noise or containing expert behavior. We test whether it successfully partitions both discrete and continuous environments. We verify in ablations that certain design choices (such as Gumbel) are important. We compare to other representation learning algorithms to demonstrate some advantages of both our state and action abstractions. Finally, we provide qualitative experiments on more complex environments that indicate our method does not fail with complex input representations and dynamics.
>
> We believe the current set of experiments are quite comprehensive, and convincingly indicate the efficacy of our representation learning approach. Our environments may be relatively simple, but are still complex enough to provide interesting results. Learning a discrete state representation end-to-end without the use of a reconstruction loss is a novel contribution. We observed a novel interesting relationship between successor representation and options. We showed the value of options which navigate to *regions* of the state space instead of specific states. We’ve provided evidence that despite the existence of degenerate minima in the SR loss, a simple max entropy regularization can avoid them.
>
> With the above perspective the additional experimentation you’ve asked for is not necessary for demonstrating the contribution, or entirely does not fit. As such, we ask you to consider increasing your respective ratings.
>
> ---
>
> We have also responded to Reviewer vZz4’s [most recent message](https://openreview.net/forum?id=Krk0Gnft2Zc&noteId=YnjXMEopLu) in-line, though our responses are meant for all reviewers. We apologize if some of the bullets there are repetitive of concepts mentioned above, but hope the additional context serves to clarify our points.

---

### Comment · Reviewer_vZz4 · 2022-11-28
**Paper not ready. Unjustified and contradictory claims.**

The authors make several claims that are either unjustified or contradict their paper. To list a few:

- "This is not a deep reinforcement learning work."
**(Contradiction)** The abstract and intro talks about how abstaction is supposed to help efficient exploration in RL. Then why shouldn't this paper demonstrate experiments on complex RL environments?

- "We now perform the abstraction end-to-end for more general environments, which is a huge advantage".
**(Unjustified)** Just proposing a new end-to-end method to replace an existing two-step method is not enough. There must be a justification why end-to-end learning is the goal. There should be some experimental validation to demonstrate that end-to-end learning indeed has some utility. In Figure 8, as authors explain, the difference comes from the action abstraction. Then, the benefit of "end-to-end learning for learning state abstraction" has not been justified.

- "We do not understand the value of environment stochasticity in this context"
**(Unjustified)** Reviewer icxB just asked for a discussion about how the method would fare in stochastic environments, which I think is a reasonable request. But all we have received are reasons why it should not be discussed.

- "Yes, discrete representations have drawbacks! Our view is that extending discrete representations to arbitrarily large environments is beyond the current scope."
**(Contradiction)** Since discrete state-action abstraction (literally the name of the proposed method) is the main contribution of this paper, there has to be an explanation or a pathway as to how the drawbacks of the discreteness could be addressed in the future.

- "It is misleading to imply that the standard in the literature is to use complex input representations. As members of this community we have often walked away from reading a paper with the belief that they operated on images, only to later find out this was not the case. Moreover, while we have provided code, most do not."
**(Contradiction)** The paper is about state abstraction. How can we not talk about complex input representations? That is the main problem that abstraction should address.

- "We are not trying to maximize an individual reward in a single environment, we are trying to come up with a good representation that generalizes to many tasks."
**(Unjustified)** Then the representation should be shown to be useful on many tasks.

- "Improving such exploration is beyond the current scope."
**(Contradiction)** In introduction, the main motivation is said to be improving exploration by abstracting states and actions.

- "Finally, even without our improvements on the state abstraction, merely proposing a new type of action abstraction is itself a contribution. Moreover we note that our experiments clearly indicate this new action abstraction is in fact better."
**(Contradiction)** The experiments do not justify state and action abstraction separately. That is why I suggest to compare to Ramesh et al (2019) + action abstraction to validate the importance of state abstraction.


- "With the above perspective the additional experimentation you’ve asked for is not necessary for demonstrating the contribution, or entirely does not fit."
I simply don't agree because of the aforementioned contradictions.
**Please reconsider additional experimentation, both to validate your contributions and scaling to complex environments, in the next iteration of this paper.**





As a final comment, it would be better to justify any claims with proper reasoning or experiments rather than making **irrefutable** statements like:
- "This is an important and valuable generalization."
- "We believe the current set of experiments are quite comprehensive, and convincingly indicate the efficacy of our representation learning approach."
- "Our environments may be relatively simple, but are still complex enough to provide interesting results."
- "it took lots of experimentation to get it right. This final success is a contribution in itself."

**Please understand that as an objective reviewer, it is hard to respond to such subjective statements.**

Because of the unjustified and contradictory claims mentioned above, I will refrain from engaging in further discussions with the authors now, unless my fellow reviewers require any further input from me.

---

> ### Author Response · Authors · 2022-11-28
> **Continuing the discussion**
>
> - (Contradiction) The abstract and intro talks about how abstraction is supposed to help efficient exploration in RL. Then why shouldn't this paper demonstrate experiments on complex RL environments?
>   - We do demonstrate more efficient exploration in our comparison experiments within the region captured by the dataset. We do not explore the rest of the environment (and neither do these competing methods).
>   - The distinction we are making when we say “not a deep RL paper” is that we do not explore the environment, we work with a dataset in that environment. We therefore compare to methods which do the same and *test on environments similar to theirs*.
> - (Unjustified) Just proposing a new end-to-end method to replace an existing two-step method is not enough.
>   - Yes, we agree that end-to-end learning is not itself a goal, but please recall our preceding comment that these “2-step” methods only apply in discrete environments. End to end learning helps us achieve this generalization.
> - (Unjustified) Reviewer icxB just asked for a discussion about how the method would fare in stochastic environments, which I think is a reasonable request.
>   - May we request references to other abstraction works which demonstrate results on stochastic environments, so that we may test on similar ones.
>   - Please recall the preceding comment which highlighted a relevant property of the SR
> - (Contradiction) Since discrete state-action abstraction (literally the name of the proposed method) is the main contribution of this paper, there has to be an explanation or a pathway as to how the drawbacks of the discreteness could be addressed in the future.
>   - We do address some future work in the conclusion, and would be more than happy to expand it based on this discussion
>   - Discrete abstraction has both drawbacks and benefits, naturally the focus of the presentation is on those benefits
> - (Contradiction) The paper is about state abstraction. How can we not talk about complex input representations?
>   - Of course this is an important topic, please recall our preceding comment that we do handle complex input representations
> - (Unjustified) The representation should be shown to be useful on many tasks.
>   - Our comparison experiments are averaged over randomly chosen tasks in the environment
> - (Contradiction) In introduction, the main motivation is said to be improving exploration by abstracting states and actions.
>   - Again, this seems like a technical misunderstanding and we apologize if our phrasing was poor. By “such exploration” we are referring to a process of expanding exploration beyond the region of the provided dataset.
> - (Contradiction) The experiments do not justify state and action abstraction separately. That is why I suggest to compare to Ramesh et al (2019) + action abstraction to validate the importance of state abstraction.
>   - We are not sure what you mean by this, our comparison to Ramesh et al. (2019) uses the same state abstraction with different action abstractions, showing that our action abstraction is an improvement
>   - Again, note that our state abstraction was learned more generally, without direct access to the underlying discrete states, and it applies to continues environments
> - As a final comment, it would be better to justify any claims with proper reasoning or experiments rather than making irrefutable statements. Please understand that as an objective reviewer, it is hard to respond to such subjective statements.
>   - We agree that such subjective statements should be avoided, and apologize for inserting them into the response.
>   - But note that you’ve taken those subjective statements out of the surrounding context which helps justify them
>   - For example, whether our environments are sufficiently complex to test our method is a subjective question (maybe it shouldn’t be, but the reality in RL is that it is). We’ve provided both “simple environment” quantitative and “complex environment” qualitative evidence (the purpose of our previous response was to list this evidence, and then yes, subjectively affirm that it is sufficient). But if you believe otherwise that is absolutely within your right.
> - Because of the unjustified and contradictory claims mentioned above, I will refrain from engaging in further discussions with the authors now, unless my fellow reviewers require any further input from me.
>   - We thank you for your time and timely response. Given the nature of your claims labeling our responses as "unjustified" or "contradictory" we would be grateful if you could consider continuing the discussion, at least to acknowledge this response. We want to remark that your review has already helped us improve and clarify our paper.
>   - The range of reviews is 3-8. That is quite large, and seems to merit further discussion.

---

### Decision · Program_Chairs · 2023-01-20

**Decision:**

Reject

**Justification For Why Not Higher Score:**

My meta-review above was quite extensive, the paper lacks a more careful empirical analysis of the proposed method and it ignores important baselines in the field.

**Justification For Why Not Lower Score:**

N/A

**Metareview: Summary, Strengths And Weaknesses:**

This paper proposes a new way of learning state and temporal abstractions with the successor representation. State abstractions are learned by clustering over the SR of a discrete representation obtained by an encoder with the Gumbel-Softmax. The temporal abstraction is obtained by learning options connecting the different clusters.

This is an interesting problem that puts forward some interesting ideas, such as using the Gumbel-Softmax to learn a discrete representation and using the SR on top of it to define clustering. One could imagine doing all sorts of things with this idea, even improving previous methods, such as Eigenoptions, by first learning a discrete representation and then computing the eigendecomposition of it. The authors also did a good job adding Ramesh’s work as a baseline, because it indeed needs to be a baseline, and in trying the proposed ideas in Montezuma’s Revenge (obtaining qualitative results is totally fine).

All that being said, the paper does fail to meaningfully evaluate the proposed idea beyond the four rooms domain. I’m generally against the trend of always asking for results in more complex domains because, when introducing an idea, it is often more important to provide intuitions and clearly demonstrate how it works in comparison to other methods, instead of desperately focusing on how to scale it up. However, in this paper some obvious experiments are missing, such as characterizing the performance of DSAA with a varying number of abstractions, this is literally why we use small domains, to be able to do this type of thorough investigation. It is also very puzzling that in the Arm2d environment DSAA was not compared to any baseline: shouldn’t at least Erraqabi et al.’s (2021) paper be used as a baseline here, given that it is the second best performing method in the four rooms domain?

Important questions about the method, such as its applicability in stochastic domains, were too quickly dismissed. It is a fair question to ask: how robust and reliable it is in face of stochasticity? The new Atari experiments (how it was written in the new version of the paper) are a little bit misleading; not because they are qualitative, but because nowhere in the paper it is said, for example, that the RAM state is what was used, by not saying it readers implicitly assume it is being done from pixels, which makes it a tad different even from the qualitative results by Machado et al. (2018) — granted that that paper looked at hundreds of options.

Because of these concerns I’m recommending the paper to be rejected. I personally really like this line of work and I recommend the authors to continue pursuing it, but I think this version of the paper needs more work. Also, I want to highlight that Erraqabi et al. (2021) evaluated their method in MuJoCo tasks, so I do think DSAA would need to be evaluated in a similar size domain if we were to contrast it to other approaches such as TACT, as much as I wrote above that I tend to be against simply asking for more domains. In this case it is quite relevant to ask how well the proposed method scales in comparison to the alternatives (the difference between TACT and DSAA is not that big in the four rooms domains).

Finally, something that I did not take into consideration in my decision, but I want to mention to the authors for them to make the paper stronger in a new version, is the work by Hoang et al. (2021). This paper also proposes abstractions based on the successor representation and it is an omission that cannot happen and it very likely should be used as a baseline as well.

Christopher Hoang, Sungryull Sohn, Jongwook Choi, Wilka Carvalho, Honglak Lee: Successor Feature Landmarks for Long-Horizon Goal-Conditioned Reinforcement Learning. NeurIPS 2021: 26963-26975

**Summary Of Ac-Reviewer Meeting:**

N/A. I sent an email to the program committee explaining that I couldn't find a suitable time not even for 2 reviewers at once.